# Deep Learning-Based Timing Offset Estimation for Deep-Sea Vertical Underwater Acoustic Communications

**Yanbo Wu** [1,2,3] , **Yan Yao** [1,2,4] , **Ning Wang** [5] **and Min Zhu** [1,2,3,*]

1   Ocean Acoustic Technology Center, Institute of Acoustics, Chinese Academy of Sciences, Beijing 100190, China; wuyanbo@mail.ioa.ac.cn (Y.W.); yaoyan@mail.ioa.ac.cn (Y.Y.)
2   Beijing Engineering Technology Research Center of Ocean Acoustic Equipment, Institute of Acoustics, Chinese Academy of Sciences, Beijing 100190, China
3   State Key Laboratory of Acoustics, Institute of Acoustics, Chinese Academy of Sciences, Beijing 100190, China
4   School of Electronic, Electrical and Communication Engineering, University of Chinese Academy of Sciences, Beijing 100049, China
5   China Research and Development Academy of Machinery Equipment, Beijing 100089, China; wn_625@163.com
*   Correspondence: zhumin@mail.ioa.ac.cn; Tel.: +86-1391-076-0416

**Abstract:** This study proposes a novel receiver structure for underwater vertical acoustic communication in which the bias in the correlation-based estimation for the timing offset is learned and then estimated by a deep neural network (DNN) to an accuracy that renders subsequent use of equalizers unnecessary. For a duration of 7 s, 15 timing offsets of the linear frequency modulation (LFM) signals obtained by the correlation were fed into the DNN. The model was based on the Pierson–Moskowitz (PM) random surface height model with a moderate wind speed and was further verified under various wind speeds and experimental waveforms. This receiver, embedded with the DNN model, demonstrated lower complexity and better performance than the adaptive equalizer-based receiver. The 5000 m depth deep-sea experimental data show the superiority of the proposed combination of DNN-based synchronization and the time-invariant equalizer.

**Keywords:** underwater acoustic communication; deep learning; timing offset estimation; human-occupied vehicle

---

## 1. Introduction

Vertical underwater acoustic (UWA) communications are critical in deep-sea activities such as scientific exploration with human-occupied vehicles. The communication waveforms are severely distorted with Doppler dilation or compression induced by the platform movement. In practical communication systems [1–5], the Doppler is usually coarsely estimated using synchronization signals, and then the residual Doppler is tracked by means of the equalizer, with the aid of pilot symbols or decided information symbols. Improving the accuracy of time-varying Doppler synchronization relaxes the burden on the adaptive equalizer and thus prevents the divergence of the adaptive algorithm in response to an impulsive noise. Among the motion sources, the fluctuation of the surface platform is the hardest to estimate because of its randomness. The ocean surface height was modeled as a sinusoid with a period of 8 s in [6], where the time difference between the transmitter and receiver was estimated over 0.5 s with the constant acceleration model. According to the experimental measurements [7] of the channel impulsive response (CIR), the surface motion was a narrow-band random process. A more accurate surface height model was described by the Pierson–Moskowitz (PM) wave height

spectrum [8], which was utilized in the analysis of the surface acoustic reflection [9]. However, the PM model has not been involved in the Doppler estimation of UWA communications.

Doppler synchronization for UWA communications consists primarily of two steps to accomplish the coarse and fine estimations, respectively [10,11]. In the first step, the time-invariant motion parameters, such as the timing offsets (displacement), the Doppler scaler (speed), and the Doppler rate (acceleration), are estimated through correlation with the synchronization signal. The correlation can be a single-branch one for Doppler-insensitive signals or a multiple-branch one [12] for Doppler-sensitive signals. The single-branch correlation has low complexity, and the estimation results are biased according to the delay–Doppler ambiguity function. The multiple-branch correlation is unbiased when enough branches are produced; however, this leads to extreme complexity in calculations. In the second step, where the continuous movement is tracked, different methods are carried out according to the modulation structure, such as using an adaptive equalizer combined with a phase-locked loop (PLL) for single-carrier (SC) systems [1] or using intercarrier inference (ICI) cancellation for orthogonal frequency division multiplexing (OFDM) systems. This paper proposes that the motion trajectory can be estimated without the tracking specified above if the relationship among the sequential synchronization results is fully utilized.

Deep learning was initially developed in the areas of computer vision and natural language processing, where mathematical description was difficult. Recently, deep learning has been prominent in wireless communications and UWA communications. The deep neural network (DNN)-based OFDM receiver proposed in [13,14] was shown to be more robust than conventional methods; this model was extended to deal with the time-varying underwater acoustic channel and was verified by simulations [7]. A DNN-based online-training UWA receiver [15] was trained by the adaptive moment estimation (Adam) optimizer for each sub-block for the time-varying UWA channel and was verified by sea trial data. To reduce the training burden, model-driven deep learning [16] used expert knowledge to make the network explainable and predictable. In [17], a DNN-based channel estimation for online UWA communications was proposed that had better simulation performance than the minimum mean squared error (MMSE) algorithm and also had less run time and storage resource consumption. In [18], a convolutional neural network was introduced to fix errors in symbol timing and carrier offset estimation in UWA communications under the condition of flat frequency response. The main difficulty of UWA communications is the high Doppler effect; we anticipate that this can be addressed by the combination of deep learning and expert knowledge to reduce training requirements.

In this paper, a novel receiver structure for underwater vertical acoustic communication is proposed, where the correlation-based estimation bias of the timing offset is learned and then estimated by the DNN to an accuracy that renders the real-time adjustment of the subsequent equalizer unnecessary. The expert knowledge of the ambiguity function of linear frequency modulation (LFM) was utilized in the design of the DNN model to reduce the parameter size for tuning. The model was based on the PM random surface height model with a moderate wind speed and was verified to be applicable for various wind speeds and experimental waveforms. This receiver, embedded with the DNN model, demonstrated lower complexity and better performance than the adaptive equalizer-based receiver. This paper's structure is as follows: Section 2 presents the channel model and the transmission packet structure. Section 3 describes the DNN model for synchronization and the full receiver used in the low-complexity baseband processing. Section 4 shows the results of the simulations and deep-sea experiments. Section 5 presents the study's conclusions.

## 2. Transmission Packet Structure and Channel Model

To reduce the synchronization complexity, we used multiple equispaced LFM signals in one packet, which is also widely used in UWA communications [19,20]. The LFM signal in the analytic signal form is written as:

$$s_{\text{Sync}}(t) = e^{j2\pi f_0 t\left(1 + \frac{f_1 - f_0}{2f_0 T_{\text{Sync}}}\right)}, \quad 0 \le t < T_{\text{Sync}}, \tag{1}$$

where $f_0$ and $f_1$ are the start and end frequencies, respectively, and $T_{\text{Sync}}$ is the duration of the synchronization signal. A whole packet consists of $N_{\text{Frm}} + 1$ LFMs, with the occurrence period of $T_{\text{Frm}}$ and the transmitted synchronization given by:

$$s(t) = \sum_{k=0}^{N_{\text{Frm}}} s_{\text{Sync}}(t - kT_{\text{Frm}}). \tag{2}$$

The $N_{\text{Frm}}$ data frames are transmitted among the synchronization signals.

The acoustic channel was modeled as the cascade of the time-invariant linear filter and the motion-induced time-varying Doppler compression or dilation. The CIR is represented by $h(t)$, and its convolution with $s(t)$ is written as:

$$r_h(t) = \int s(t - \tau)h(\tau)d\tau. \tag{3}$$

The received waveform in the passband is then given by:

$$r_{\text{PB}}(t) = r_h\left(t - \frac{d(t)}{c}\right) + w(t), \tag{4}$$

where $d(t)$ is the time-varying distance between the transmitter and the receiver, $c$ is the speed of the sound, and $w(t)$ is the additive noise. During the vertical communication between the submersible and the mother ship, the position of the mother ship rapidly fluctuates with the surface wave height. Therefore, in this scenario, the displacement is modeled by the time-varying surface wave height. The Pierson–Moskowitz model was adopted, whose spectrum is given in [8] as:

$$\psi_{\text{SW}}(f) = \frac{2\pi\alpha g^2}{(2\pi f)^5}e^{-\beta\left(\frac{g}{2\pi f U}\right)^4}, \tag{5}$$

where $U$ is the wind speed, and the other constant parameters take the following values: $\alpha = 0.0081$, $\beta = 0.74$, and $g = 9.81$.

## 3. DNN-Based Synchronization and the Proposed Receiver

Although the DNN model can learn from many of the received samples and can estimate the desired time-varying Doppler, the large number of parameters prevents its application if the DNN's input is the whole-packet waveform or the DNN's output is time-varying time offsets. Two pieces of expert knowledge were used to reduce the dimension of the DNN for synchronization: the ambiguity function of the LFM and the spline reconstruction of the motion displacement, whose functional modules were integrated before and after the DNN, respectively. Using the DNN for the timing offset bias correction, the receiver was then fully rebuilt with lower complexity and better performance than the traditional adaptive equalizer.

### 3.1. Input/Output of the DNN Model and Its Training Method

The coarse timing offset was estimated by cross-correlation between the received LFM and the local one. However, its bias was significant because of the uncompensated random movement. The delay bias, $\Delta$, of the LFM is a determined function of the Doppler scaler, $\gamma$, written as follows:

$$\Delta = -\frac{T_{\text{Sync}}\gamma}{(f_1 - f_0)(1 - \gamma)}\left(\frac{f_1}{1 - \gamma} - \frac{f_0\gamma}{1 + \gamma}\right). \tag{6}$$

The derivative of $d(t)$ is the time-varying speed, written as $v(t) = \partial d(t)/\partial t$. The timing offset and the Doppler scaler are written as $\tau(t) = d(t)/c$ and $\gamma(t) = v(t)/c$, respectively.

According to the packet structure, the timing offsets $\{\tau_B(kT_{Frm})\}_{k=0}^{N_{Frm}}$ at the start position of the synchronization signals are estimated by the correlation; however, they have the Doppler-induced bias, written as:

$$\tau_B(kT_{Frm}) = \tau(kT_{Frm}) + \Delta(kT_{Frm}), \qquad 0 \leq k \leq N_{Frm}, \tag{7}$$

where, for a small Doppler scaler, $\gamma(kT_{Frm}) \ll 1$, the bias is approximated by:

$$\Delta(kT_{Frm}) = -\rho\gamma(kT_{Frm}), \tag{8}$$

and the delay–Doppler ratio is written as:

$$\rho = -\frac{f_1}{f_1 - f_0}T_{Sync}. \tag{9}$$

To learn the correlation among the timing offsets, the DNN was fed with the biased timing offsets in one packet with a fixed input scaler. Furthermore, to improve the numerical efficiency, the desired output of the DNN was the scaled bias vector rather than the direct timing offsets. The training diagram for the DNN timing bias estimation is shown in Figure 1. The biases generated by Equation (7) with the scaler and the DNN output were compared in the DNN optimizer.

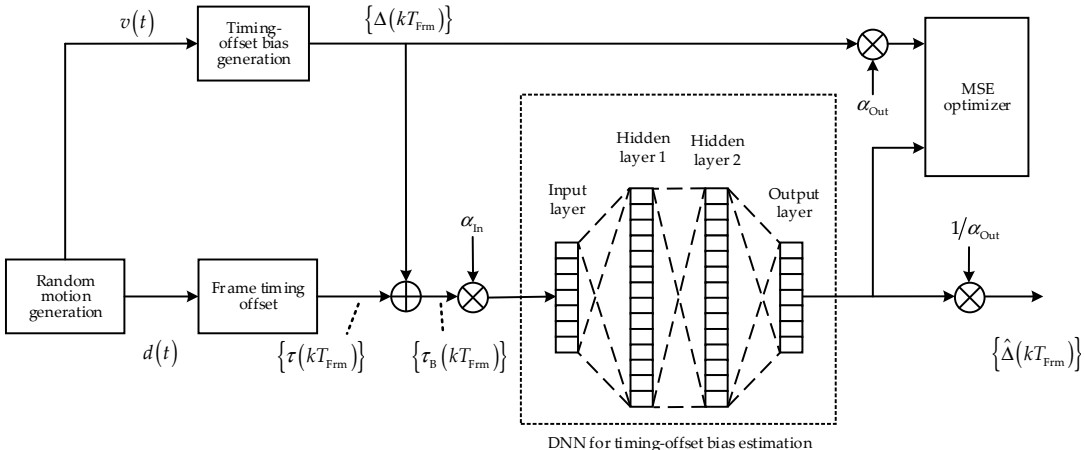

**Figure 1.** Deep neural network (DNN) training diagram for the timing bias estimation. MSE, mean square error.

### 3.2. Internal Structure of the DNN Model

The DNN consists of one input layer, $L$, hidden layers, and one output layer. We used the vectors $\{\mathbf{y}_l\}_{l=0}^{L}$ to represent the value of the layers. The input layer vector is calculated using the biased timing offsets enlarged by a fixed scaler, $\alpha_{In}$, written as:

$$\mathbf{y}_0 = \alpha_{In}\begin{bmatrix} \tau_B(0) & \tau_B(T_{Frm}) & \dots & \tau_B(N_{Frm}T_{Frm}) \end{bmatrix}^T. \tag{10}$$

Similarly, the output vector is the multiplication of the output scaler, $\alpha_{Out}$, and the expected bias of the timing offsets, written as:

$$\mathbf{y}_L = \alpha_{Out}\begin{bmatrix} \Delta(0) & \Delta(T_{Frm}) & \dots & \Delta(N_{Frm}T_{Frm}) \end{bmatrix}^T. \tag{11}$$

The generation of the internal layers can be described as follows:

$$\mathbf{y}_l = f_l(\mathbf{w}_l\mathbf{y}_{l-1} + \mathbf{c}_l), \quad 0 \leq l < L, \tag{12}$$

where $\mathbf{w}_l$ and $\mathbf{c}_l$ are the multiplication matrix and the additive vector for the linear transformation, respectively, and $f_l(*)$ is the activation function for the nonlinear transformation. The activation function was selected as the rectified linear unit (ReLU) for the internal layers, whose expression is given by:

$$f_l(x) = \max(0, x). \tag{13}$$

The output layer is directly obtained by linear transformation without the activation function, written as:

$$\mathbf{y}_L = \mathbf{w}_L \mathbf{y}_{L-1} + \mathbf{c}_L. \tag{14}$$

The size of the internal layers and the scalers $\alpha_{\mathrm{In}}$, $\alpha_{\mathrm{Out}}$ are the hyper-parameters defined before the DNN's optimization. Under the given hyper-parameters, the parameters $\{\mathbf{w}_l; \mathbf{c}_l\}_{l=0}^{L}$ are optimized by minimizing the mean square error (MSE).

The random displacement $d(t)$ is generated by the surface wave model in Equation (4). The DNN is trained with simulation data instead of the experimental data to avoid overfitting.

### 3.3. Full Receiver in Baseband

The proposed synchronization was based on the model-driven DNN. The core algorithm in the proposed synchronization, which was built by the DNN, was used to suppress the correlation peak ambiguity between the delay and the Doppler; the other parts in the receiver used classical methods to reduce the dimension of the learning parameters. As a result of the accurate compensation of the time-varying relative motion, the subsequent equalizer did not need to be time-invariant. The receiver structure is shown in Figure 2 and described in detail as follows.

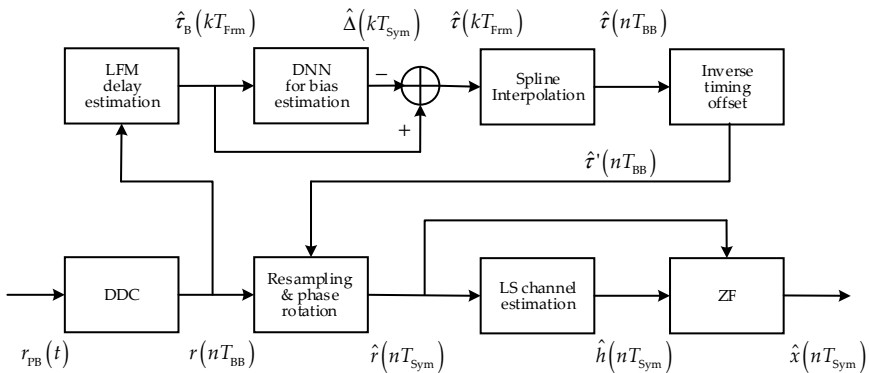

**Figure 2.** Receiver structure. LFM, linear frequency modulation; DDC, digital down converter; LS, least-squared; ZF, zero forcing equalization.

The first step was to convert the passband waveform $r_{\mathrm{PB}}(t)$ into the baseband one $r[n]$ with a low sampling rate, $T_{\mathrm{BB}}$. Conversion in accordance with the digital down converter (DDC) is given as:

$$r[n] = \mathrm{LPF}\left\{ r_{\mathrm{PB}}(mT_{\mathrm{PB}}) e^{-j2\pi m T_{\mathrm{PB}} f_{\mathrm{C}}} \right\}\Big|_{n=\frac{mT_{\mathrm{PB}}}{T_{\mathrm{BB}}}}, \tag{15}$$

where $\mathrm{LPF}\{\cdot\}$ is the low pass filter, $f_{\mathrm{C}}$ is the carrier frequency, and $T_{\mathrm{PB}}$ is the passband sampling interval of the DDC input.

For the timing offset estimation, we first obtained the discrete timing offsets at the start of the synchronization signals using traditional correlation. Then, the bias vector was learned from the DNN and subtracted. Finally, the continuous timing offsets were interpolated by a cubic spline function.

For the *k*-th synchronization signal in the received waveform, correlation to the baseband waveform is carried out as:

$$z_k[n] = \left| \sum_m r[m + n + kT_{\text{Frm}}] s^*_{\text{Sync}}(mT_{\text{BB}}) e^{j2\pi f_c(mT_{\text{BB}} - kT_{\text{Frm}})} \right|. \tag{16}$$

The integer timing offset is written as:

$$\tau_{\text{I}, k} = \underset{n}{\text{argmax}}\{z_k[n]\}, \tag{17}$$

and then the fractional offset is acquired by maximizing the parabolic fitting curve around it as follows:

$$\tau_{\text{F}, k} = \frac{z_k[\tau_{\text{I}, k} + 1] - z_k[\tau_{\text{I}, k} - 1]}{4z_k[\tau_{\text{I}, k}] - 2z_k[\tau_{\text{I}, k} - 1] - 2z_k[\tau_{\text{I}, k} + 1]}. \tag{18}$$

The timing offset estimate with the bias is as follows:

$$\hat{\tau}_{\text{B}}(kT_{\text{Frm}}) = (\tau_{\text{I}, k} + \tau_{\text{F}, k})T_{\text{BB}}. \tag{19}$$

The biased timing offset vector in the whole packet is fed into the DNN, and the bias vector $\{\hat{\Delta}(kT_{\text{Frm}})\}$ is then obtained, which is eliminated to get the unbiased estimation, as follows:

$$\hat{\tau}(kT_{\text{Frm}}) = \hat{\tau}_{\text{B}}(kT_{\text{Frm}}) - \hat{\Delta}(kT_{\text{Frm}}), \qquad 0 \le k \le N_{\text{Frm}}. \tag{20}$$

Finally, the timing offset for each sample in the packet is obtained by cubic spline interpolation from the timing offsets at the synchronization signals:

$$\hat{\tau}(nT_{\text{BB}}) = \text{Spline}\left\{\{(kT_{\text{Frm}}, \hat{\tau}(kT_{\text{Frm}}))\}_{k=0}^{N_{\text{Frm}}}; nT_{\text{BB}}\right\}, \tag{21}$$

where Spline{·} is the cubic spline interpolation function in the "not-a-knot" condition.

The timing offset compensation was carried out to recover the transmitted signal without the motion effects. As shown in Figure 3, the compensation in the low sample-rate baseband form included three steps: the inversion of the timing offset, the Farrow filtering, and the carrier offset ration compensation. The timing offset curve was resampled to meet the requirement of the Farrow filter's input being equispaced. Assuming that the inverse time offset is $\{\hat{\tau}'(nT_{\text{BB}})\}_n$, the time transformation in the resampling compensation described as $\{(nT_{\text{BB}}, nT_{\text{BB}} - \hat{\tau}'(nT_{\text{BB}}))\}_n$ is the inverse function of that with the estimation $\{(mT_{\text{BB}}, mT_{\text{BB}} - \hat{\tau}(mT_{\text{BB}}))\}_m$. Therefore, the inverse timing offset can be obtained as follows:

$$\hat{\tau}'(nT_{\text{BB}}) = nT_{\text{BB}} - \text{Linear}\left\{\{(mT_{\text{BB}} - \hat{\tau}(mT_{\text{BB}}), mT_{\text{BB}})\}_m; nT_{\text{BB}}\right\}. \tag{22}$$

It is further used for resampling by Farrow filter and phase rotation, written as:

$$r'(nT_{\text{BB}}) = \text{Farrow}\left\{\{(mT_{\text{BB}}, r[m])\}_m; \ nT_{\text{BB}} - \hat{\tau}'(nT_{\text{BB}})\right\} e^{-j2\pi f_c \hat{\tau}'(nT_{\text{BB}})}, \tag{23}$$

where the first item on the right side is the Farrow filter's output with the delay adjustment $\hat{\tau}'(nT_{\text{BB}})$. As shown in Figure 3, the Farrow filter consists of $N_{\text{F}} + 1$ component filters $\{f_k(n)\}_{k=0}^{N_{\text{F}}}$ and the output of Farrow filtering is the polynomial function of $\hat{\tau}'(nT_{\text{BB}})$ and the filter array outputs. As the sample clock is well recovered, the resampled waveform is directly decimated into the symbol rate for the following equalization.

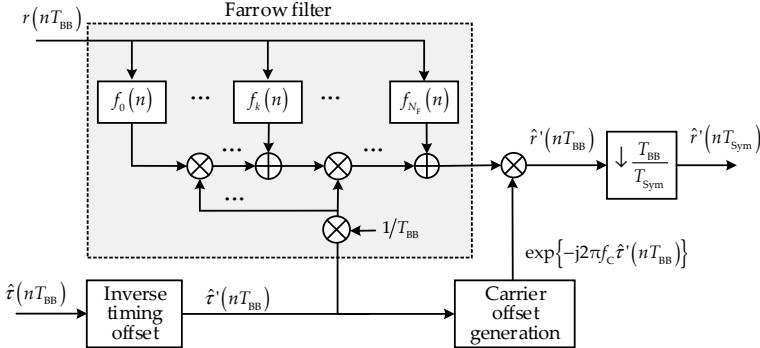

**Figure 3.** The timing offset compensation in the baseband.

The resampled waveform $r'(nT_{\mathrm{Sym}})$ is assumed to be well compensated for in the motion effects, and the residual CIR is time-invariant. As shown in Figure 2, the CIR is obtained as $\left\{\hat{h}(nT_{\mathrm{Sym}})\right\}$ for each frame based on least-squared (LS) channel estimation, and then the zero-forced equalization is carried out to obtain the output symbols $\left\{\hat{x}(nT_{\mathrm{Sym}})\right\}$.

## 4. Simulation and Experimental Results

The start and end frequencies of the LFM synchronization signal were $f_0 = 7.5$ kHz and $f_1 = 12.5$ kHz, respectively, and its duration was $T_{\mathrm{Sync}} = 25.6$ ms. Therefore, according to (8), the ratio of the bias delay to the Doppler was $\rho = -0.064$ s. The duration of the quadrature phase shift keying (QPSK) symbol was $T_{\mathrm{Sym}} = 0.2$ ms and the passband and baseband sample intervals were $T_{\mathrm{PB}} = 0.0125$ ms and $T_{\mathrm{BB}} = 0.1$ ms, respectively. For example, with a Doppler scaler $\gamma = 0.001$, the delay bias after the correlation was $\Delta = \rho\gamma = 0.064$ ms, which caused a symbol phase rotation of 115.2º, and therefore the adaption of the equalizer was necessary if the bias was uncompensated. In the packet, there were $N_{\mathrm{Frm}} = 14$ frames whose individual duration was $T_{\mathrm{Frm}} = 0.4928$ s, and the total number of the LFM signals was $N_{\mathrm{Frm}} + 1 = 15$, which was equal to the sizes for both the input and output of the DNN model. Table 1 outlines the hyper-parameters of the DNN model. The model was trained in the simulation data generated in accordance with Figure 1, with a constant wind speed of 15 m/s. The number of training samples was 100,000, with the epochs being 50. The training was completed in 56 s on a laptop computer with an Intel i7-8750 CPU at a 2.2 GHz clock frequency.

The bias was estimated by the trained DNN model and eliminated in the delay estimation. The unbiased positions of the LFMs were interpolated by the cubic spline function to obtain the timing offset for the whole packet. For comparison, the unbiased positions were interpolated by the linear function in the simulations. The additional positions used include biased positions directly given by the correlation and perfect positions. The number of the validation samples was 2000 for each wind speed condition, and the wind speed in the test varied from 5 to 25 m/s. The root-mean-square errors (RMSEs) of the timing offsets for the whole packet were compared in the simulations, as shown in Figure 4. It should be noted that we first investigated the performance of the DNN models trained at different wind speeds to select the appropriate DNN model, and the results are shown in Figure 4 for comparison. It can be concluded that the model trained at a specific wind speed can cope with all test scenarios at lower wind speeds and some test scenarios at higher wind speeds within a certain range, since the spectral component of high wind speed can cover the spectral component of low wind speed. However, for extremely high wind speeds, the sample distribution space is wider, and more training samples are needed to avoid overfitting. Based on the above factors, in the following analysis, the network model trained at a wind speed of 15 m/s was adopted to process the test data. We then focused on the performance comparison of the pairwise combination of the three LFM position estimation methods and the two interpolation methods. The biased LFM synchronization performed the worst for both interpolation methods, and because of the bias causing RMSE degradation, spline interpolation did not improve the performance as compared to linear interpolation. After the bias was eliminated through

the DNN model, it performed nearly the same as the perfect LFM synchronizations under the condition of either interpolation method, and specifically, spline interpolation could obtain a half-RMSE of linear interpolation. For the proposed scheme of the DNN model and spline interpolation, the RSME was nearly 0.01 ms in various wind speeds, which means a phase rotation error of 18°. The RMSE floor was mainly caused by the high-frequency components of the surface motion and the limited repetition frequency of the LFMs. Increasing the LFMs for the same packet duration suppressed the RMSE but sacrificed the transmission bandwidth.

**Table 1.** Hyper-parameters of deep learning models.

| Parameters | Value |
|---|---|
| Scaler of input vector | 1000 |
| Scaler of output vector | 0.0001 |
| Size of input layer | 15 |
| Size of hidden layers | {128, 128} |
| Size of output layer | 15 |
| Activation function of hidden layers | ReLU |
| Activation function of output layer | None |
| Loss function | MSE |
| Learning rate | 0.001 |
| Batch size | 128 |
| Optimizer | Adam |
| Total number of parameters | 20,495 |

Note: ReLU is rectified linear unit and Adam is adaptive moment estimation.

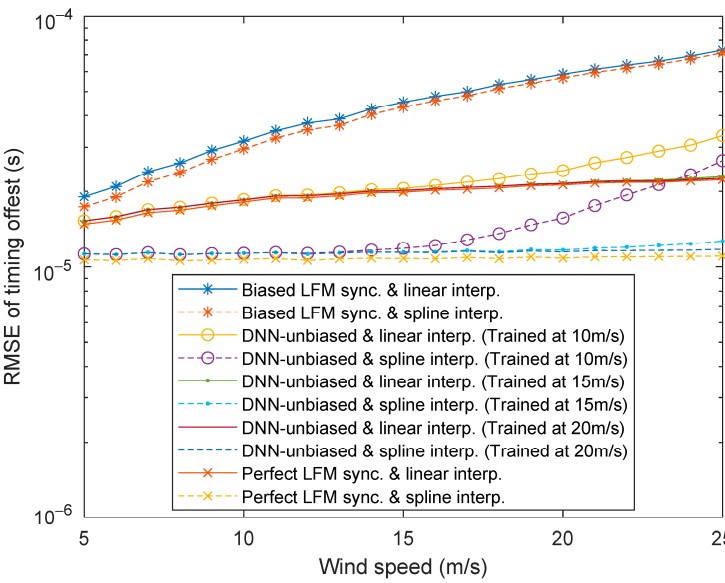

**Figure 4.** The timing offset estimation performances in the simulations. RMSE is root-mean-square error.

The computational complexities of the main functional models for one packet are depicted in Table 2. The total complexity for the proposed scheme was approximately six million multiply–accumulate operations (MACs), which is 24% of that of a traditional scheme. It can be seen that the complexity of the DNN was negligible because of the usage of the expert knowledge of the timing estimation. The model with the most operations in the proposed scheme was the resampling model, which was

also halved as a result of using the integer symbol-spaced sampling rate rather than the fractional one. The complexity of the equalizer was tremendously reduced because of the time-invariant structure.

**Table 2.** Complexities of the receivers for one packet.

| Functional Model | Proposed Scheme (MAC) | Adaptive Equalizer Scheme (MAC) |
|---|---|---|
| LFM cross-correlation | 0.77 M | 0.77 M |
| DNN for bias estimation | 20 k | - |
| Spline interpolation | 0.17 k | - |
| Resampling | 4.9 M | 9.9 M |
| LS channel estimation | 4.0 k | - |
| ZF equalization | 0.28 M | - |
| Adaptive equalization | - | 14 M |
| Total | 6 M | 25 M |

Note: MAC, multiply–accumulate operation.

The proposed DNN-based scheme was verified in the experimental data packet, which was sampled in the 2011 sea trials of China's first deep manned submersible "Jiaolong." The experimental condition was introduced by Zhu [1]. Unfortunately, the wind speed of the experiments was not recorded. However, from the timing estimation, it can be deduced that the surface wave height from the valley to the peak was 2.25 m and the period was approximately 8 s; these measurements can be used for accurate descriptions of the instant sea condition. The vertical and horizontal communication distances were 5030 m and 2390 m, respectively. The packet consisted of 15 LFMs and 14 data frames, and for each frame, the numbers of the training symbols and the information symbols were 200 and 1936, respectively. Only the internal 10 frames were used in the performance comparisons because the two boundary frames on each side had fewer LFM signals nearby for synchronization. During the experiments, the receiver consisted of four hydrophone channels, which were jointly processed using the model from [1]. The hydrophone in the first channel was omnidirectional, and the other three hydrophones were cone-directional; their signal noise ratios (SNRs) were 9.3, 20.1, 18.5, and 16.9 dB, respectively and the total SNR was 18.9 dB (as described in [1]), which was calculated using four-channel equal-gain combining. In this paper's comparisons of the traditional and proposed methods, we only used the second channel, which had the largest SNR, and the adaptive equalizer was replaced by the time-invariant one for the robustness of the impulsive noise. Figure 5 illustrates the timing offset before unbiasing and after interpolation, as well as the bias output of the DNN, which are marked in the previous section as $\hat{\tau}_B(kT_{Frm})$, $\hat{\tau}(nT_{BB})$, and $\hat{\Delta}(kT_{Frm})$, respectively. We can see that, after spline interpolation, the timing offset was much smoother, which fits the true movement, and the bias estimated by the DNN was in proportion to the first-order derivative of the displacement.

As the perfect timing offsets were not available, the DNN-based synchronization methods and the biased LFM synchronization methods were compared in the QPSK phase trajectories and symbol error rates (SERs), as shown in Figure 6. As seen in Figure 6a, for the bias LFM synchronization and linear interpolation, the symbols in the middle of the frame usually endured the maximum offset because of the uncompensated acceleration effect, which was first analyzed by Sharif [21]. The spline interpolation could suppress the acceleration, and thus the error of the phase trajectory was linearly changing between the boundaries of each frame (see Figure 6b) and the phase error was significant at the end of the frame because of the uncompensated synchronization bias. After the bias was learned from the DNN model, the QPSK symbols were well recovered, as shown in Figure 6d, with an SER of 0.002. This was considered an excellent result for the single-channel received scheme of LFM-based synchronization followed by a simple time-invariant equalizer.

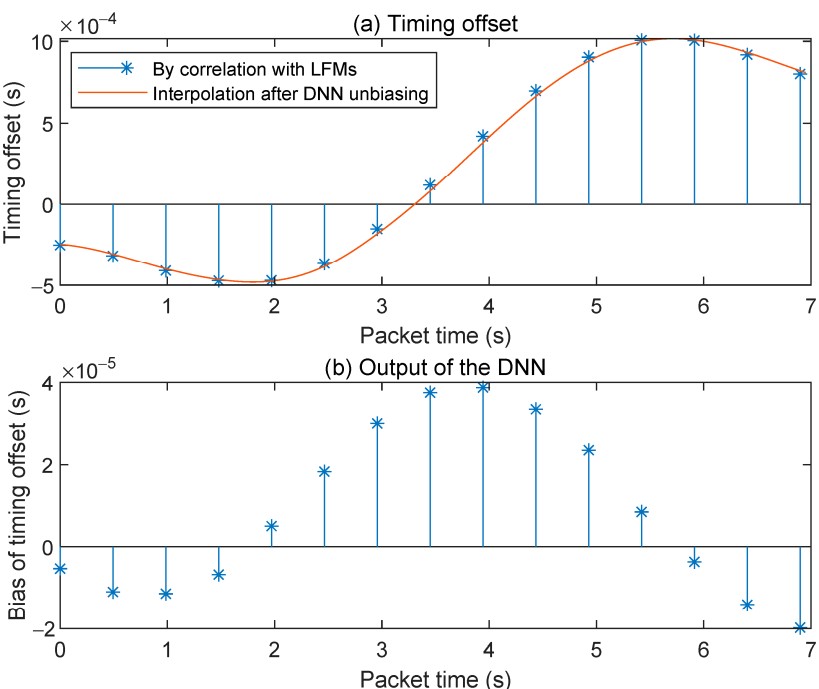

**Figure 5.** Timing offsets before and after interpolation (**a**) and the output of the DNN (**b**) in the vertical communication experiment.

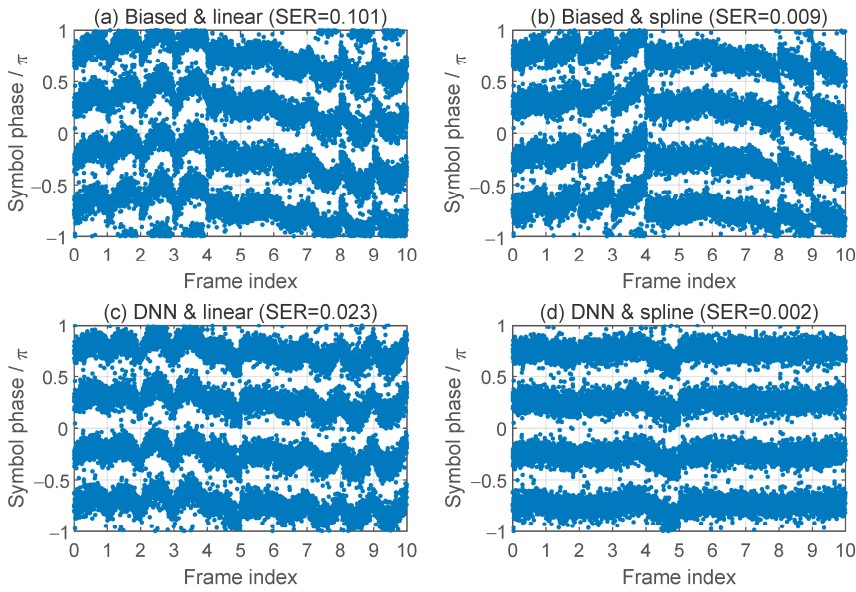

**Figure 6.** QPSK phase trajectories and symbol error rates (SERs) in the experimental verification (**a**) Biased LFM with linear interpolation (**b**) Biased LFM with spline interpolation (**c**) DNN-unbiased with linear interpolation (**d**) DNN-unbiased with spline interpolation.

## 5. Conclusions

The rapid variation of the underwater acoustic vertical channel was estimated successfully with the proposed DNN-based synchronization method. To decrease the learning burden, the DNN was model-driven and embedded with expert knowledge. For a duration of 7 s, 15 timing offsets obtained at the start of the LFMs with correlation bias were fed into the DNN. Although the DNN model was trained with the surface wave height generator at fixed wind speed, the DNN model showed its robustness in response to a large range of wind speeds and experimental waveforms. Combined with

spline interpolation to simulate real movement, the timing offset compensation could make the channel time-invariant. The successful time-invariant equalization in the symbol-rate sampling using 5000 m depth deep-sea experimental data showed the superiority of the proposed DNN-based synchronization.

**Author Contributions:** Conceptualization, Y.W. and M.Z.; methodology, Y.W. and N.W.; software, Y.W., Y.Y., and N.W.; investigation, Y.W. and M.Z.; writing—original draft preparation, Y.W. and Y.Y.; writing—review and editing, Y.Y. and M.Z.; supervision, M.Z. All authors have read and agreed to the published version of the manuscript.

**Funding:** This work was financially supported by the National Natural Science Foundation of China (Grant Nos. 61971472 and 61471351), the Strategic Priority Research Program of the Chinese Academy of Sciences (Grant No. XDA22030101), the National Key Research and Development Program of China (Grant No. 2016YFC0300300), and the China Scholarship Council (Grant No. 201904910081).

**Conflicts of Interest:** The authors declare no conflict of interest.

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
