# Peer review of "Deep Learning-Based Timing Offset Estimation for Deep-Sea Vertical Underwater Acoustic Communications"

_applsci, doi:10.3390/app10238651_

Round 1
Reviewer 1 Report
The article deals with an important problem of accurate synchronization of a communication signal in a time-varying underwater acoustic channel. I read it with a greate pleasure. The authors presented both the problem and the proposed innovative solution in a clear and detailed way.
I have very few comments:
1. Please remove the repeated paragraph in Chapter 4, from the words: "The proposed DNN-based ..." to "... first-order derivative of the desplacement."
2. Just before Equation 14: I think that the Digital Down Converter should be written in capital letters if the abbreviation (DDC) is given immediately afterwards.
3. Under table 1 You wrote: "Different from the scheme of four-channel combination [1], in this paper, we only use the second channel which had the largest SNR (...)". What was this SNR value? And what was it for other channels used in experiment described in [1]?
I also have one question about implementation issues: have you tried to estimate whether such a DNN network needs more, less or a similar amount of computing resources to operate (without the learning process) than the adaptive equalizer combined whith a phase-locked loop?
Reviewer 2 Report
This work addresses the issue of surface platform motion (due to wind-driven ocean surface waves) in the scenario of deep-sea vertical underwater acoustic communications. The authors use a deep-learning based neural network, trained with a surface wave model (based on Pierson-Moskowitz spectrum), to improve the performance of acoustic communication, in particular, the accuracy of Doppler estimation in the receiver. The results from simulation and sea experimental data show that the combination of deep neural network and cubic spline interpolation yields a highly accurate estimation, so that no further fine estimation process is needed. This leads to a simplification of the receiver structure, as previous structures would generally require both coarse and fine Doppler estimations. The proposed work is original. The described methods look correct to me. The manuscript is well written. Therefore, I recommend accepting this paper for publication in Applied Sciences.
Minor Comments:
The Pierson-Moskowitz model, mentioned in the abstract and the introduction, is not explicitly described in Section 2 when introducing Eq. 4. I suggest clarifying it for readers.
The sea surface condition (e.g., wind speed, sea state) of the experiment is not clearly described in this paper, neither in the reference [1]. Depending on the actual sea condition, the results reported here can be even more (or less) significant.
Large winds generate low-frequency, but high-amplitude ocean waves. In other words, for wind-driven ocean surface waves, the wave period and amplitude both increase with the wind speed. Can the authors comment why the network trained with a moderate wind speed could work for a variety of wind speeds, which yield different heights and periods? Furthermore, will the network trained with a variety of wind speed yield an even better performance?
